# In-Situ Vascular Regeneration by Host Cells of Acellular Human Saphenous Vein Implanted in Porcine Carotid Artery

**DOI:** 10.3390/ijms26104718

**Published:** 2025-05-15

**Authors:** Andrew Bond, Vito Bruno, Nadiah Sulaiman, Jason Johnson, Sarah George, Raimondo Ascione

**Affiliations:** 1Translational Health Sciences, Bristol Medical School, University of Bristol, Bristol Royal Infirmary, Bristol BS2 8HW, UK; andrew.bond@bristol.ac.uk (A.B.); vito.d.bruno@bristol.ac.uk (V.B.); nadiahsulaiman@ukm.edu.my (N.S.); jason.l.johnson@bristol.ac.uk (J.J.); s.j.george@bristol.ac.uk (S.G.); 2Department of Tissue Engineering and Regenerative Medicine, Faculty of Medicine, Universiti Kebangsaan Malaysia, Cheras, Kuala Lumpur 56000, Malaysia

**Keywords:** endothelial-like cells, endothelium, cell seeding, vascular graft, bioengineering, vein graft failure

## Abstract

Small vascular graft engineering may help reduce early vein graft failure. We assessed the feasibility, safety, and in vivo vascular regeneration potential of the decellularised human saphenous vein (D-hSV) with and without pre-seeding with porcine endothelial-like cells (ELCs) following grafting in porcine carotid artery (CA). A total of 14 pigs received CA grafting of control D-hSVs (n = 7) or D-hSVs seeded with ELCs (SD-hSV; n = 7). Ultrasound vascular Doppler was undertaken before and after grafting, and at 4 weeks. Outcome measures included patency, intimal thickening (IT), in situ vascular regeneration, endothelial cell (EC) coverage, neo-angiogenesis, mesenchymal–EC transition, and contractile cells. All animals reached the predefined culling point in good health, with no feasibility/safety concerns. Mild graft dilatation occurred at 4 weeks vs. baseline, with no difference between groups. In total, 9/14 grafts (64.3%) remained patent at 4 weeks (4/7 (57.1%) vs. 5/7 (71.4%) in the D-hSV and SD-hSV groups, respectively). IT increased from 17.1 ± 4.7% at baseline to 54.1 ± 12.2% at 4 weeks. Vascular regeneration occurred in all patent grafts with EC coverage, an increase in collagen and elastin, vimentin, SM-MHC-11, and calponin, with no difference between groups. The D-hSV for arterial vascular grafting is feasible and safe and associated with signs of in situ vascular regeneration by host cells at 4 weeks. Pre-seeding with ELCs did not add benefits.

## 1. Introduction

Severe atherosclerotic disease leads to myocardial infarction (MI) and limb ischemia/amputation globally [1]. These patients benefit from coronary artery (CABG) or peripheral artery bypass grafting (PABG), which involve using predominantly autologous saphenous vein (SV) grafts [2,3]. However, the patency rate of SV grafts (SVGs) is only 50–60% at 10 years for CABG patients, and 25% at 10 years for PABG patients, due to thrombosis and excessive intimal hyperplasia [4,5]. SV harvesting/grafting triggers 76–100% of endothelial cell (EC) damage and loss of viability, depending on the storage and handling conditions [6], predisposing patients to vein graft failure [7]. SVGs are often not available due to their poor quality or previous stripping. Concomitantly, about 20% of all bypass grafts performed during CABG surgery involve using the internal mammary artery, leading to excellent patency rates of 92–95% at 15–20 years [8,9,10]. For PABG patients, however, there are no arterial graft options, with artificial conduits made from expanded polytetrafluoroethylene (ePTFE) [2] or Dacron (polyester fabric), representing the only alternatives to SVGs, despite only 67% and 49% of patency at 2 and 5 years [2], high infection rates, thrombogenicity, pseudo-aneurysms, calcification, and poor suturability. To address these issues, new vascular engineering technologies are being developed using biological or synthetic scaffolds functionalised with cells and/or molecules [11]. Pre-seeding ECs on these scaffolds has been attempted to prevent thrombosis and for nitric oxide release to reduce smooth muscle cell migration within the vascular wall [12] and associated intimal thickening (IT) [13,14]. However, pre-seeding with endothelial colony-forming cells (ECFCs) derived from blood cells has shown controversial results in preclinical studies [15,16], suggesting that cell pre-seeding may not be beneficial. Previously, we showed the feasibility of implanting decellularised human SVs (D-hSVs) in porcine carotid arteries without immunosuppression [17] or a plain thin-layered nanotextile small arterial graft made from biodegradable polymers [18], with evidence of in situ arterialisation by host cells in both cases at 4 weeks post-implant. In addition, we reported a simple method to isolate porcine endothelial-like cells (pELCs) from porcine blood and seed them onto D-hSVs [19]. The aim of the present study was to assess the feasibility and safety of implanting D-hSV grafts with or without pre-seeded pELCs in porcine carotid arteries and the occurrence of signs of in situ vascular regeneration.

## 2. Results

All animals reached the predefined termination point (29.4 ± 0.4 days) without complications. Nutrition, well-being, and weight gain were within reference intervals. The weight at 4 weeks was 82.9 ± 2.6 kg and 80.0 ± 1.3 kg in the D-hSV and SD-hSV groups, respectively (*p* = 0.35), representing weight gains of 22.5 ± 1.7% and 26.8 ± 2.0% in the SD-hSV and D-hSV groups, respectively (*p* = 0.13) (Table A1). All grafts were easy to suture with no evidence of tearing, bleeding, rupture, or haematoma. At 4 weeks, there was no evidence of infection, aneurysm formation, or neoplastic degeneration. Overall, 5/14 grafts (35.7%) were occluded with luminal thrombus. Descriptive data for occluded grafts are shown in Table A2. Staining with Martius scarlet blue (MSB) was undertaken to detect luminal thrombus (Figure A2) as this identifies red blood cells (RBCs), collagen, and fibrin/muscle. Occluded grafts showed blockage by complex thrombus, with RBCs, intertwined with fibrin/muscle cells and collagen (Figure A2c). Patent grafts only showed some RBCs in the lumen, likely deposited at the time of termination (Figure A2d), or thin-mixed RBCs and fibrin patches over the intima. Qualitatively, there were no differences in RBC deposition between SD-hSV or D-hSV grafts (Figure A2a and Figure A2b, respectively).

### 2.1. Microbiology and Haematology

All grafts were confirmed sterile before grafting, with a lack of anaerobic bacterial growth in the media and with cells seeded onto grafts being mycoplasma-free. Haematology screening was obtained in 10/14 pigs. All variables remained within reference intervals at 4 weeks and did not differ between groups (Table A3). The neutrophil/lymphocyte ratio (NLR) and the leukocyte/monocyte ratio (LMR) did not change over time in both groups.

### 2.2. Graft Patency, Luminal Remodelling, and Intimal Thickening

At 4 weeks, 9/14 (64.3%) of all grafts remained patent with the patency rate of SD-hSV grafts being 71.4% (5/7) vs. 57.1% (4/7) in the D-hSV group (*p* = 0.58). In vivo ultrasound vascular Doppler (USD) data of the nine patent grafts are shown in Table 1 and Table 2, with no differences between the groups. Similarly, inner wall diameters and the diameter matching ratio (DMR) of patent grafts did not differ between groups, with only the trends for D-hSV grafts showing some outward remodelling (mean DMR > 1), whereas SD-hSV grafts showed remodelling more inwardly (mean DMR < 1). At 4 weeks, the peak blood flow velocity (PBFV) in the proximal (Px.RCA) portion of D-hSVs was significantly reduced compared to post-implant (*p* < 0.05), while both PBFV and diastolic (DBFV) were significantly reduced in the distal (D.RCA) of SD-hSV grafts (*p* < 0.05). Other vascular data did not differ, as shown in Figure A3 and Figure A4 and Table A4 and Table A5.

Histological analysis revealed that the average inner diameter of all 14 native RCAs was 1.6 ± 0.1 mm (1.8 ± 0.3 mm and 1.4 ± 0.1 mm for the D-hSV (n = 7) and SD-SV (n = 7) groups, respectively (*p* = 0.3)). The pre-grafting inner diameter of all 14 D-hSVs was 2.8 ± 0.1 mm with no difference between the groups. Elastin content within pre-grafting RCAs and untouched LCAs at termination was 57.9 ± 3.9% vs. 51.3 ± 4.4% (*p* > 0.05) (Table A6). As expected, the elastin content of pre-grafting D-hSV grafts (22.7 ± 3.1%), was much lower than the pre-grafting RCA, with no differences between the groups. We have previously shown that the protocol used to derive D-hSVs in vitro is associated with no presence of live residual cells [17]. The pre-grafting coverage with endothelial cells obtained with the ELC seeding of SD-hSV grafts was 61.3 ± 11.3%, although when reassessing after surgery, for the SD-hSV specimens leftover from 5/7 grafts, the luminal surface coverage with endothelial cells was lower at 48.3 ± 8.9%. Of note, the coverage with endothelial cells of RCA segments excised before grafting was 96.7 ± 1.1%. Histological samples at 4 weeks were not fixed under physiological pressure; hence they were subjected to shrinkage, preventing meaningful comparisons with in vivo USD vascular data. Histology confirmed pre-termination USD data, with 9/14 grafts being patent. Intimal thickening was present across the proximal and distal segments of the grafts at 4 weeks (Table 3). This was 54.1 ± 12.2% and 49.2 ± 10.1% for the proximal and distal segments of patent grafts, respectively, with a trend for less intimal thickening between patent D-hSV vs. SD-hSV grafts. No IT was observed at 4 weeks within the native RCA or the untouched left carotid arteries.

Vessel remodelling at 4 weeks is shown in Figure 1a–d. The inner wall diameter of patent grafts increased to 5.4 ± 0.8 mm and 5.4 ± 0.9 mm (proximal and distal graft regions, respectively; both *p* < 0.05 vs. native RCA) and 4.8 ± 0.7 mm and 3.7 ± 0.6 mm for the proximal and distal regions, respectively, of occluded grafts; both *p* < 0.05 vs. native RCA (Figure 1d). Data subdivided by control versus seeded vein are shown in Figure A5. At the same time point, the overall diameter of the LCA as an in situ arterial control was 2.0 ± 0.1 mm vs. 1.5 ± 0.1 mm for pigs receiving either D-hSV (n = 4) or SD-hSV (n = 7) grafts, respectively.

A large fibrous capsule was seen surrounding all grafts (Figure A6a) with no differences in size between patent and occluded grafts or between D-hSV and SD-hSV groups (Figure A6b,c; *p* > 0.05).

### 2.3. In Situ Vascular Regeneration by Host Cells

At 4 weeks, all nine patent grafts showed substantial coverage of the luminal surface with CD31-positive endothelial cells (Figure 2). This was higher at the proximal and distal ends of the grafts at the interface with the native RCA and less, but still tangible, in the middle of the grafts (Figure 2a). No difference in endothelial cell coverage was seen between groups (Figure 2b).

H&E staining and nuclear counterstains at 4 weeks showed marked but quantitatively similar intramural cell infiltration in all grafts (11/12 and 8/12 proximal and distal graft regions) across groups and regardless of patent or occluded condition (Figure 3a,d,e). Patent grafts contained large amounts of elastin, collagen, and fibrin (Figure 3b,c,e) and the mesenchymal–endothelial transition marker (vimentin) in the intima and perivascular layers (Figure 3d and Figure 4).

The qualitative evaluation of contractile proteins, calponin, and smooth muscle myosin heavy chain-11 (SM-MHC) indicated the presence of contractile smooth muscle cells in the middle portion of all patent grafts, predominantly within the thickened intima (representative images of an SD-hSV and D-hSV (Figure 4)), with no differences between groups. The presence of these markers was higher in the Px and D. anastomotic regions and very high in the control Px.RCA and D.RCA as expected (Figure 4).

While the elastin content in all grafts was higher at 4 weeks vs. baseline grafts, the abundance was lower than the native porcine RCA or LCA (Figure 5 and Table A6). This difference applied to both patent grafts (2.0 ± 1.4 and 10.1 ± 5.6%; Px. and D.graft, respectively) and occluded grafts (6.2 ± 3.7% and 16.1 ± 7.3%; Px. vs. D.graft, respectively). The elastin content differed across the regions of each graft (*p* < 0.05) and was consistently higher in the Px.RCA vs. D.RCA (*p* < 0.05), including when the analysis focused only on patent grafts. Cell seeding did not affect elastin content vs. unseeded controls.

Extensive adventitial neo-angionesis was detected at 4 weeks in all grafts, regardless of patent or occluded and D-hSV or SD-hSV status (Figure A7). Representative examples to demonstrate the co-localisation of DBA (Dolichos Biflorus Agglutinin)-lectin and CD31 in ECs around the patent graft lumens in the same graft are shown in Figure 2e,f and Figure A7a,b,d,e. The sections of the LCA as in situ arterial control at the same time point are shown as positive staining controls for complete endothelium of CD31 and DBA-lectin (Figure 2c and Figure 2d, respectively).

## 3. Discussion

This study supports the concepts of feasibility, safety, and in situ vascular regeneration when implanting acellular D-hSVs in the pig carotid artery. While there were no feasibility/safety concerns related to suturability, tearing, haematology, aneurysmal, or neoplastic degeneration, 64% of the implanted grafts were patent and showed clear signs of in situ vascular regeneration by host cells. This study also suggests that pre-seeding the acellular graft with pELCs had no beneficial effects.

The 64% patency rate is interesting when considering the xenotransplantation aspect, the lack of immunosuppression, and only 48% of luminal coverage achieved by cell pre-seeding. This may reflect the fact that the human veins were decellularised before implantation and the daily use of 300 mg of Aspirin, a dosage in line with clinical guidelines [20,21]. Noticeably, implanting vein grafts with limited luminal coverage by endothelial cells is typical of the current surgical practice, as the marked loss of endothelial cells is associated with harvesting/handling SVs for CABG/PABG surgery [22], along with reduced cell viability [23], triggering more effective methods of SV harvesting [24] and of endothelial preservation [25]. The lack of immunosuppression in this study is in keeping with the established clinical practice of implanting commercial patches or artificial biological heart valves made from decellularised/acellular bovine pericardium in humans every year [26].

A key finding was the occurrence of signs of in situ vascular regeneration by host cells, including extensive re-endothelialization (expression of CD31 and DBA-lectin), intramural recellularization with mesenchymal–endothelial transition cells (vimentin), smooth muscle cells/SMCs (contractile proteins SM-MHC-11, calponin), and adventitial neo-angiogenesis. This was combined with marked extracellular matrix remodelling (elastin and collagen) and the pulsatility index. The luminal surface coverage with ECs was dense at the proximal and distal parts of the grafts, suggesting migration across the anastomoses from the adjacent RCAs. A large number of ECs were also observed in the mid segment, possibly migrating from the RCAs or resulting from circulating progenitors. While we can be sure that 100% of ECs seen in the control D-hSV group were host cells, this cannot be confirmed for the SD-hSV group, as those grafts had an average pre-seeding coverage of 48% before implantation. Future studies could use cell labelling approaches (e.g., green fluorescent protein (GFP) [27], or implanting male donor cells into female recipients and performing fluorescent in situ hybridization [28]) to determine whether CD31-positive cells at termination are host-derived or residual seeded cells. Quint et al. [27] decellularised tissue-engineered grafts grown from banked porcine smooth muscle cells and then re-endothelialised the obtained extracellular matrix with endothelial progenitor cells (EPCs) or endothelial cells from the intended porcine recipient, achieving 65% lumen coverage before implantation into the porcine carotid artery. In total, 3/3 of their nude grafts were occluded, while 5/5 of the re-endothelialised grafts were patent at 30 days. For 4/5 of these patent grafts, the endothelial coverage was made up of 35% of the seeded GFP-labelled cells, whereas the remaining 65% consisted of host cells. Our study differs markedly from the work by Quint and colleagues regarding graft preparation, and this might explain the different outcome related to the nude graft, with our study showing >50% of nude grafts being patent at 30 days, with extensive re-endothelialization by host cells. Yet, both studies appear to indicate that pre-seeding with ECs may not be beneficial for vascular tissue engineering, as most if not all of the cells present in the graft after in vivo exposure appear to be host cells. This is in keeping with relevant findings by others, suggesting that monocytes seeded onto a mouse venous tissue-engineered graft rapidly disappear within 2 h after implant in vivo [29].

A large amount of transmural SMCs were also observed at 4 weeks with an abundance of SM-MHC-11 and calponin proteins. Again, this was more pronounced at the proximal and distal ends vs. the mid portion of the grafts, although the number of SMCs observed was lower than that seen in the native RCA. While we can speculate that these SMCs migrated across the anastomoses from the native RCA or were derived from circulating progenitors, more work is needed to clarify their origin. Based on their anatomical position within the wall of the graft, it is unlikely that these SMCs could have migrated from the adventitial layer, given that most of them appear to be located within the neointimal region and inside the D-hSV layer (Figure 4). In this same region, there was also an abundance of vimentin, regarded as a mesenchymal–endothelial transition marker. This might indicate that both contractile and synthetic SMCs were in phenotypic flux, explaining why some contractile markers were also present.

Exposing veins to the higher shear stress of arterial circulation (10–70 dyn/cm^2^ in arteries vs. 1–6 dyn/cm^2^ in veins [30,31]) and higher partial oxygen pressure leads to intimal thickening [32]. Adaptation to these dramatic changes may be of utmost importance to maintain graft patency. The USD data of this study on the inner wall diameter at 4 weeks showed an occurrence of outward remodelling with a 57% increase in inner wall diameter. This was not uniform, as the inner wall diameter increased by 11% in the proximal segments of patent grafts, decreased by 2.7% in the proximal segments of occluded grafts, and increased by 35% in the distal segments of patent grafts, becoming similar to pre-implanted RCA segments. The increase in inner wall diameter at USD at 4 weeks was reflected by a tangible increase in elastin content. Elastin is needed in arterial vessels to provide the required elasticity/pulsatility during the cardiac cycle and contribute to the PI level, which reflects the resistance in a pulsatile arterial system [33]. In this study, the average PIs for the porcine RCA prior to surgical excision and the LCA at termination were 4.3–4.9 and 2.7–3.9, respectively. Noticeably, implanted grafts showing a PI < 5 soon after implantation remained patent at 4 weeks, while those with a PI > 5 soon after implantation were occluded at 4 weeks, while no differences in the PI were seen between patent D-hSV and SD-hSV grafts (Table 2). These differences might simply reflect baseline differences in D-hSV quality and wall thickness.

Five grafts were occluded at 4 weeks. Early vein graft thrombosis is associated with suboptimal surgical techniques, a poor quality of conduit [34,35], and graft-to-artery mismatch [36] or poor run-off. To minimise the rate of early thrombosis, surgery was undertaken by senior heart surgeons on large pigs (60 kg) to gain good-sized porcine RCAs with good size matching with the human D-hSV grafts. However, the implanted D-hSVs were derived from SV segments discarded at human surgery and therefore not of the ideal quality. This factor might have contributed to the observed 5/14 graft occlusions observed in this study, in keeping with the notion that up to 12% of SV grafts are considered too diseased at CABG surgery, leading to a 50% occlusion rate compared to non-diseased SV grafts [37]. Alternatively, these occlusions might be due to the xenotransplantation aspect of implanting human SV grafts in pigs. However, the decellularization method used has shown no residual viable cells on the D-hSV grafts [17], and 9/14 grafts were patent at 4 weeks despite xenotransplantation. A fibrous capsule was observed at 4 weeks in all grafts with infiltrating inflammatory cells and lymphoid-like nodules. This could indicate the presence of residual human epitopes on the implanted grafts, possibly triggering rejection. However, in a parallel study using autologous porcine venous grafts in the same RCA porcine model, we did not observe this adventitial fibrous capsule (Figure A8). This may suggest that the presence of a fibrous capsule is unlikely in a clinical research setting designed to use autologous D-hSV.

This study has limitations. It might be argued that the sample size was small. However, this was a feasibility/safety pilot study; therefore, with no sample size calculation, we included seven animals in each group, which is in keeping with the large animal sample sizes used by others [27]. In relation to the observed signs of in situ vascular regeneration by host cells, the follow-up period of 4 weeks might be regarded as too short. Keeping the animals for longer might have helped ascertain if the level of in situ vascular regeneration by host cells is time-dependent. In addition, the carotid artery interposition graft model used is not fully relevant to the CABG/PABG surgical practice. In addition, all pigs used in this study were disease-free. Based on the findings from this study, future studies are planned, and for added translational benefit, could potentially additionally use hypertensive or diabetic pig models. Finally, an additional limitation of our study is that we used only female pigs and therefore this study may not account for sex-based differences in vascular biology, immune response, and vascular remodelling typical of male pigs. However, we used only female pigs to minimise intra-experimental variability to increase the robustness of the findings while minimising animal husbandry issues associated with using a mixed gender cohort of animals [38].

## 4. Materials and Methods

Methods are extensively shown in Section A.1. All data underlying this manuscript are included here or in the online Appendix A. Full raw data can be made available on request.

### 4.1. Ethics

D-hSVs were produced in the lab as by-products from segments of human SVs left over from patients undergoing CABG procedures under approval by the local Research Ethics Committee (Ref. 10/H0170/63). Donors gave generic consent for research use as part of the donation process and are not identifiable. This study conformed to the Declaration of Helsinki principles. The animal procedures were conducted at the MHRA-compliant Translational Biomedical Research Centre (TBRC, Bristol, UK) in line with UK Home Office regulations (Animal Act 1986) and the guidelines from Directive 2010/63/EU under Project Licences 30/2854 and 30/3064, granted by the Home Office after approval by the local Animal Welfare and Ethics Review Body (AWERB). Reporting of research data was in keeping with the ARRIVE (Animals in Research: Reporting In Vivo Experiments) guidelines [39].

### 4.2. Human Saphenous Vein Decellularisation

Human SVs (n = 10) were collected under sterile conditions, stored in saline at 4 °C, and subsequently decellularised as previously described [17]. D-hSVs were stored in Dulbecco’s phosphate-buffered saline (DPBS; Sigma-Aldrich, Gillingham, UK) at 4 °C until use. Of the 10 veins collected, four were long enough to be divided into two segments and used for two implants in pigs (Table A1), giving a total of 14 grafts implanted.

### 4.3. Porcine Endothelial-like Cell (ELC) Isolation and Seeding

ELCs were derived from pig blood (n = 3) as previously described [19]. Briefly, peripheral blood mononuclear cells (PBMCs) were isolated from porcine blood using density gradient separation with Ficoll Paque Plus 1077 (density 1.077 g/mL; GE-Healthcare, Hatfield, UK) in Sepmate50 tubes (Stemcell Technologies, Cambridge, UK). Monocytes were allowed to adhere onto culture plates coated with type-I rat tail collagen (Gibco, Waltham, MA, USA) and cultured in ELC media (endothelial cell growth medium-2 Bulletkit (Lonza, Cambridge, UK; CC-3162), including additional foetal bovine serum (FBS, heat inactivated, Gibco) to a final concentration of 11%), until growth of ELCs. ELCs were characterised by a clear cobblestone morphology and the co-expression of the endothelial cell markers CD31, VE-cadherin, and von Willebrand factor via immunocytochemistry. After expansion, cells were cryopreserved in CellBanker 2 (Amsbio, Abingdon, UK) until required. Two weeks prior to surgery, cells were thawed, expanded in 75 cm^2^ flasks (Corning, Glendale, AZ, USA) at 37 °C/5% CO_2_ for one week in ELC media. Cell seeding is reported in Section A.1.1. Briefly, one week before surgery, ELCs were enzymatically detached from flasks at 37 °C, tested/counted for viability, and pipetted into the lumen of the selected D-hSV segment, with the distal end tied with a 4-0 prolene suture, at 1 × 10^6^ ELC per 1 cm of graft, suspended in 50–100 µL of ELC media. The proximal end was then tied with a suture. Seeded D-hSVs (SD-hSVs) were then placed in 3 mL of ELC media in a 5 mL vented cap tube, on a roller at 1 rpm (37 °C, 5% CO_2_) for 1.5 h, then the tubes containing tissue were inverted and placed back on the roller to enhance the uniform seeding of cells across the conduit. The following day, the inversion process was repeated, and media replenished. After 96 h, the ties were removed, and the resulting SD-hSVs were placed into 5 mL pre-warmed ELC media under static conditions until implantation. On day 7 post-seeding, small samples from the ends of D-hSVs (n = 7) and SD-hSVs (n = 7) were removed and fixed in 10% (*v*/*v*) formalin to be used as pre-grafting controls, while the mid segments were kept in ELC media at 37 °C and taken to the TBRC facility for surgical implantation. The time from the initiation of seeding to surgical grafting was approximately 140 h.

### 4.4. In Vivo Surgical Grafting and Vascular Doppler Acquisition

D-hSV grafts with (SD-hSV: n = 7) or without (D-hSV: n = 7) pre-seeded ELCs were randomly and blindly implanted in the right carotid artery (RCA) position in female landrace large white pigs (n = 14; weight 61.2 ± 1.0 kg; age 3.9 ± 0.0 months—Table A1) under general anaesthesia (GA), as previously reported [17]. Animals received 75 mg of Aspirin with food daily during the 5 days before surgery, followed by 300 mg of Aspirin daily from day 1 after surgery until termination. No immunosuppression was used. A skin incision was made at the right neck to expose the RCA. Heparin was administered to achieve an activated clotting time >250 s. Following clamping, a 0.5 cm segment of the mid RCA was excised (fixed in 10% (*v*/*v*) formalin as histology control), and 1.5–2.0 cm long D-hSV or SD-hSV grafts were interposed using 7-0 Prolene monofilament running sutures. Vascular clamps were removed, and grafts were spread externally with papaverine (30 mg/mL, Torbay Pharmaceuticals, Paignton, UK) to prevent spasm. Ultrasound vascular Doppler (USD) imaging (Edge II ultrasound machine, with HFL38xi 13-6 MHz linear transducer; Fujifilm SonoSite, Bothell, WA, USA) was acquired of the native mid RCA before grafting and of the proximal RCA, implanted grafts, and distal RCA 15 min and 4 weeks after grafting (Figure A1).

### 4.5. Outcomes Measures

#### 4.5.1. Microbiology

To assess the microbiological contamination of the implanted grafts, 100 µL of remaining media was plated onto sheep blood agar (Oxoid, Basingstoke, UK) and incubated at 37 °C, 5% CO_2_ for 24 h. Plates were then assessed for visible colonies indicating contamination. In addition, ELC culture media was tested for the presence of mycoplasma using a PCR Mycoplasma Test Kit, following the manufacturer’s protocols (Promocell, Heidelberg, Germany).

#### 4.5.2. Haematology

Four mL of venous blood was collected into vacutainer EDTA K2 tubes (1.8 mg/mL; Becton Dickinson, UK) at baseline and at 4 weeks post-implant and stored at 4 °C. Haematological screening was conducted by a commercial diagnostics lab (Langford Vets Diagnostic Laboratories, Bristol, UK). NLR and LMR were calculated as indicators of systemic inflammation and vein graft failure, respectively [40].

#### 4.5.3. Feasibility/Safety Measures

At surgical implant and 4 weeks, grafts were assessed for suturability, tearing, bleeding, rupture, and haematoma via direct vision and by USD. Histology was then carried out to test for infection, graft occlusion (luminal thrombus formation), excessive dilatation, aneurysm formation, and neoplastic degeneration.

#### 4.5.4. Patency, Intimal Thickening, and Vascular Regeneration

Graft patency was assessed via USD and histology, while intimal thickening (IT) and signs of vascular regeneration were assessed via immunohistochemistry (luminal coverage with endothelial cells (CD31 and DBA–lectin), extracellular matrix remodelling (elastin, collagen), the presence of intramural mesenchymal–endothelial cell transition (vimentin), contractile proteins (SM-MHC-11, calponin) and neo-angiogenesis in the adventitial area (see Methods in Appendix A)). USD measures included DMR as a ratio of the inner diameter of the proximal (Px.)RCA or distal (D.)RCA to the graft to determine mismatch between the graft and native vessel. Graft patency was defined as the persistence of a patent lumen at pre-termination. We also assessed the length of the graft at implant, blood flow velocity (via pulse wave Doppler; PWD), systolic/peak (PBFV) and diastolic (DBFV) flow velocities, and the pulsatility index (PI = (PBFV − DBFV)/Time-averaged flow velocity) as markers of vessel/graft elasticity.

### 4.6. Statistical Analysis

To test for a statistically significant difference in outcome measures between seeded and non-seeded grafts, a chi-square test was used. Comparisons of Doppler ultrasound data between time points (implant vs. explant), or pig graft outcomes (patent vs. occluded) were performed using an independent samples t-test. When analysing graft data, due to heterogeneity along the h-SVs, each conduit was treated as an individual entity due to the small number of paired samples. Differences between regions were determined using either an ANOVA, or univariate analysis of variance, and post hoc Tukey tests. Differences were deemed significant when the *p*-value was less than 0.05. IBM SPSS Statistics (Version 26) was used for all analyses.

## 5. Conclusions

In conclusion, the use of D-hSVs for vascular grafting in arterial position is feasible and safe and is associated with signs of in situ vascular regeneration by host cells at 4 weeks. Pre-seeding with porcine ELCs was not effective. More studies are warranted with a focus on long-term outcomes.

## Figures and Tables

**Figure 1 ijms-26-04718-f001:**
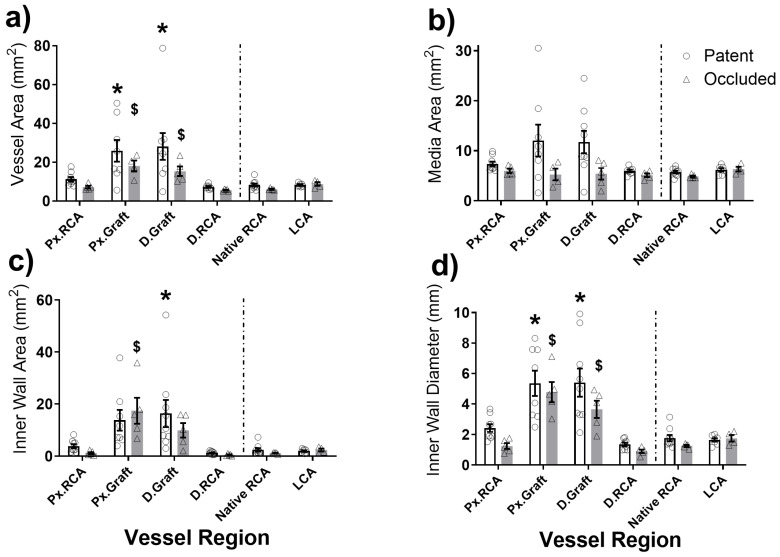
Histological measurement of (**a**) vessel area, (**b**) medial area, (**c**) inner wall area, and (**d**) inner wall diameter of segments across patent (white bars) or occluded (grey bars) grafts (all grafts included—not split by pre-seeding). Pre-implant native RCA and explant LCA measurements are also included for comparison. RCA—right carotid artery, Px.—proximal, D.—distal, LCA—left carotid artery as control. *p* < 0.05 vs. Native RCA for patent (*) and occluded ($) grafts.

**Figure 2 ijms-26-04718-f002:**
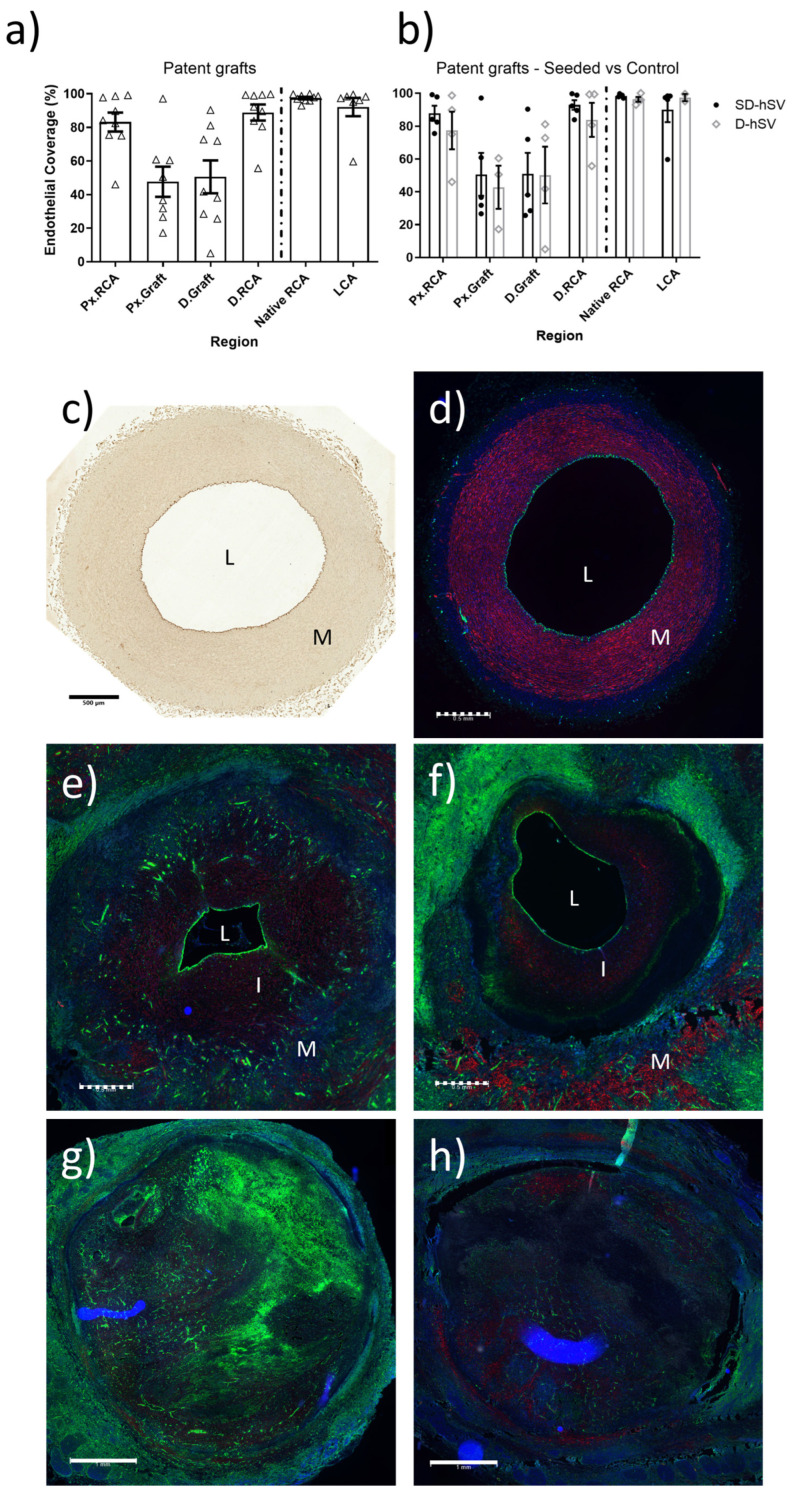
Coverage of luminal surface with endothelial cells (ECs; %) of regions along vascular conduit of (**a**) all patent grafts and (**b**) patent grafts subdivided by seeded (SD-hSV) or control (D-hSV). (**c**) CD31 (dark brown) immunohistochemistry and (**d**) DBA (Dolichos Biflorus Agglutinin)-lectin (green) staining of a left carotid artery showing ECs in the lumen. (**e**–**h**) Representative images of DBA-lectin-stained SD-hSV (**e**,**g**) and D-hSV (**f**,**h**) graft sections. Representative patent (**e**,**f**) and occluded (**g**,**h**) grafts are shown. DBA-lectin (green), α-smooth muscle actin (red), and DAPI (nuclei, blue). L—Lumen, I—intimal thickening, and M—media. Scale bars represent 500 μm (black bar; (**c**), and white dashed line; (**d**–**f**)) and 1000 μm (white solid line).

**Figure 3 ijms-26-04718-f003:**
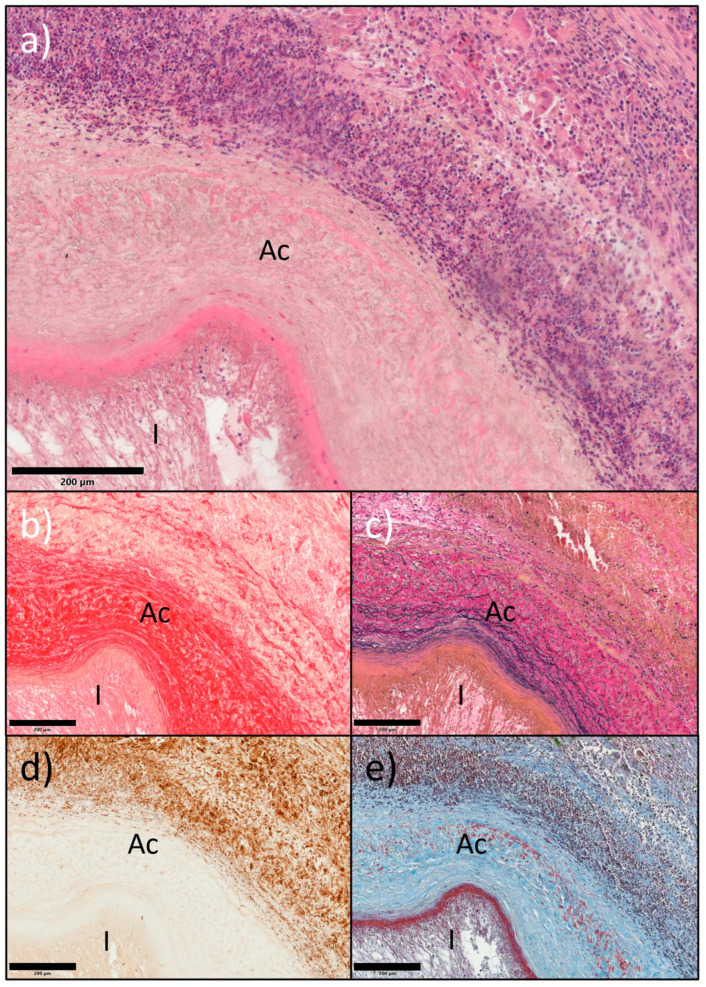
Adjacent sections of a region of an SD-hSV, patent at termination, containing an acellular area (Ac), stained for (**a**) H&E (nuclei, purple), (**b**) picrosirius red (collagen, red), (**c**) elastin van Gieson (elastin, purple), (**d**) immunohistochemistry of vimentin (brown, nuclei, blue), and (**e**) Martius scarlet blue (MSB) (fibrin, red; collagen, blue; nuclei, brown; red blood cells, green). I—intimal thickening. Scale bars represent 200 μm.

**Figure 4 ijms-26-04718-f004:**
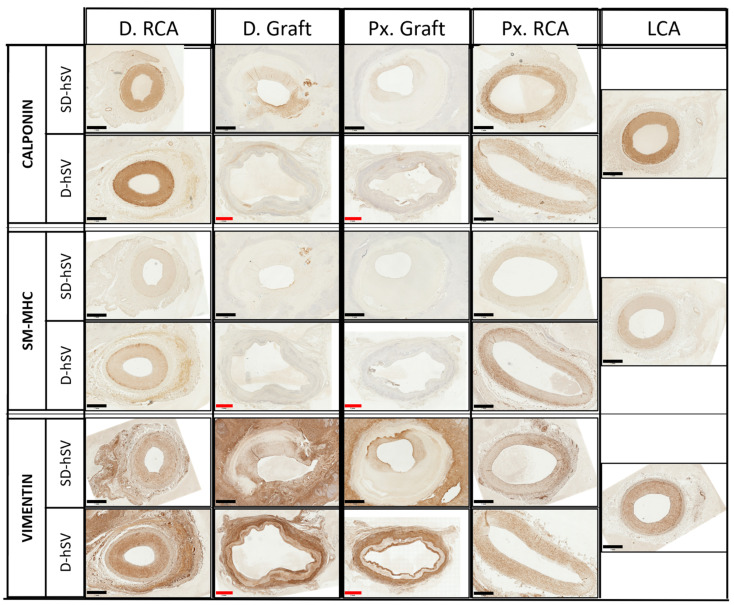
Representative immunohistochemical staining of unseeded control (D-hSV and cell-seeded (SD-hSV) grafts). Immunohistochemical staining (marker of interest, brown) of the contractile smooth muscle cell markers calponin and smooth muscle myosin heavy chain 11 (SM-MHC) and vimentin (mesenchymal–endothelial transition cells), at regions along an explanted seeded (SD-hSV) and unseeded (D-hSV) graft and in an un-grafted left carotid artery (LCA) control. D—distal; Px—proximal. Scale bars are 1 mm (black) or 2 mm (red).

**Figure 5 ijms-26-04718-f005:**
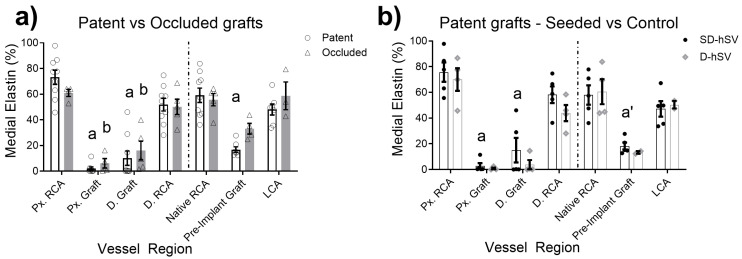
Elastin content (%) across graft regions of (**a**) patent vs. occluded and (**b**) control (D-hSV) vs. ELC-seeded (SD-hSV) patent grafts. RCA—right carotid artery, LCA—left carotid artery, Px.—proximal, and D.—distal. Significance levels: a—*p* < 0.01 compared to native RCA patent; b—*p* < 0.01 compared to native RCA occluded; a’—*p* < 0.05 SD-hSV compared to native RCA.

**Table 1 ijms-26-04718-t001:** Inner wall diameters and diameter matching index of control vs. seeded patent grafts.

	Region	Pre-Implant		15 min Post-Implant		4 Weeks Post-Implant	
		D-hSV (n = 4)	SD-hSV (n = 5)	D-hSV	SD-hSV	D-hSV	SD-hSV
**Inner Wall Diameter (cm)**	**Mid RCA**	0.48 ± 0.05	0.44 ± 0.03				
	**Px.RCA**			0.37 ± 0.03 (n = 4)	0.39 ± 0.04 (n = 4)	0.44 ± 0.05 (n = 4)	0.40 ± 0.03 (n = 5)
	**Graft**			0.36 ± 0.04 (n = 4)	0.38 ± 0.04 (n = 5)	0.45 ± 0.16 (n = 4)	0.68 ± 0.16 (n = 5)
	**D.RCA**			0.29 ± 0.05 (n = 4)	0.38 ± 0.05 (n = 5)	0.45 ± 0.05 (n = 4)	0.47 ± 0.02 (n = 5)
	**Mid LCA**					0.53 ± 0.02 (n = 2)	0.51 ± 0.02 (n = 5)
**Diameter Matching Ratio (DMR)**	**Px.RCA/Graft**			1.08 ± 0.14 (n = 4)	1.18 ± 0.22 (n = 4)	1.52 ± 0.62 (n = 4)	0.81 ± 0.25 (n = 5)
	**D.RCA/Graft**			0.79 ± 0.08 (n = 4)	1.10 ± 0.19 (n = 5)	1.51 ± 0.60 (n = 4)	0.92 ± 0.29 (n = 5)

Vascular Doppler data. Means ± SEM. n = number of pigs imaged. D-hSV: control grafts; SD-hSV: seeded grafts.

**Table 2 ijms-26-04718-t002:** Blood flow measurements in control vs. seeded patent grafts by vascular Doppler.

	Region	Pre-Implant		15 min Post-Implant		4 Weeks Post-Implant	
		D-hSV	SD-hSV	D-hSV	SD-hSV	D-hSV	SD-hSV
**Peak Blood Flow Velocity (cm/s)**	**Mid RCA**	151.0 ± 16.9 (n = 4)	175.5 ± 39.2 (n = 5)				
	**Px.RCA**			81.1 ± 11.7 ^a^ (n = 4)	48.9 ± 12.6 (n = 5)	43.3 ± 7.1 ^a^ (n = 4)	55.4 ± 12.3 (n = 5)
	**Graft**			184.9 ± 66.5 (n = 4)	143.8 ± 49.2 (n = 5)	225.7 ± 70.2 (n = 4)	226.3 ± 91.7 (n = 5)
	**D.RCA**			186.6 ± 33.8 (n = 4)	184.1 ± 33.2 ^a^ (n = 5)	151.6 ± 75.9 (n = 4)	84.1 ± 15.2 ^a^ (n = 5)
	**Mid LCA**					163.6 ± 47.8 (n = 2)	149.7 ± 17.5 (n = 5)
**Diastolic Blood Flow Velocity (cm/s)**	**Mid RCA**	35.0 ± 5.9 (n = 4)	69.1 ± 27.6 (n = 5)				
	**Px.RCA**			33.9 ± 6.6 (n = 4)	21.0 ± 5.0 (n = 5)	18.1 ± 2.1 (n = 4)	19.6 ± 4.2 (n = 5)
	**Graft**			59.9 ± 23.0 (n = 4)	50.2 ± 14.6 (n = 5)	79.6 ± 27.0 (n = 4)	74.1 ± 33.6 (n = 5)
	**D.RCA**			77.9 ± 14.2 (n = 4)	64.0 ± 11.2 ^a^ (n = 5)	42.9 ± 20.7 (n = 4)	23.1 ± 5.1 ^a^ (n = 5)
	**Mid LCA**					37.5 ± 12.4 (n = 2)	36.6 ± 6.6 (n = 5)
**Pulsatility Index**	**Mid RCA**	3.04 ± 0.70 * (n = 4)	5.61 ± 0.66 * (n = 4)				
	**Px.RCA**			1.59 ± 0.53 (n = 4)	2.04 ± 0.43 (n = 4)	8.28 ± 5.99 (n = 4)	3.12 ± 0.61 (n = 5)
	**Graft**			2.25 ± 0.28 (n = 4)	2.23 ± 0.64 (n = 4)	2.75 ± 0.39 (n = 4)	3.43 ± 0.64 (n = 5)
	**D.RCA**			1.55 ± 0.25 (n = 4)	2.22 ± 0.26 (n = 4)	2.30 ± 0.36 (n = 4)	25.88 ± 23.06 (n = 5)
	**Mid LCA**					3.72 ± 0.05 (n = 2)	3.96 ± 0.85 (n = 5)

Means ± SEM. n = number of pigs imaged. * *p* < 0.05 indicates significance between control and seeded grafts; ^a^ *p* < 0.05 indicates significance between conduit at implant vs. explant. D-hSV: control grafts; SD-hSV: seeded grafts.

**Table 3 ijms-26-04718-t003:** Degree of intimal thickening in vein grafts prior to implant, and at the proximal and distal grafts at explant.

		D-hSV	Proximal Graft	Distal Graft
	**All (n = 14)**	15.8 ± 3.5%	71.4 ± 9.7%	67.1 ± 9.2%
	**Patent (n = 9)**	17.1 ± 4.7%	54.1 ± 12.2%	49.2 ± 10.1%
	**Occluded (n = 5)**	12.8 ± 4.7% ^a^	99.0 ± 0.8%	99.4 ± 0.6%
**Control** **n = 7**	**All**		63.4 ± 17.6%	60.2 ± 15.7%
	**Patent**		28.5 ± 18.4%	31.2 ± 14.2%
	**Occluded**		98.3 ± 1.1%	98.9 ± 1.1%
**Seeded** **n = 7**	**All**		78.2 ± 10.4%	74.1 ± 10.3%
	**Patent**		69.5 ± 12.6%	63.7 ± 11.3%
	**Occluded**		100.0 ± 0.0%	100.0 ± 0.0%

^a^: n = 4 for occluded D-hSV; mean ± SEM.

## Data Availability

All datasets presented in this manuscript can be made available upon reasonable request.

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
