# Peer review of "In-Situ Vascular Regeneration by Host Cells of Acellular Human Saphenous Vein Implanted in Porcine Carotid Artery"

_ijms, 2025, doi:10.3390/ijms26104718_

Round 1
Reviewer 1 Report
Comments and Suggestions for Authors
This a very interesting study and the findings add to the knowledge of heterologous vascular grafting. There is a very large amount of data and some of the table are difficult and hard to understand, specially when they showed no difference between both groups. it would be very helpful for the reader to show it in a more friendly manner.
Author Response
Reviewer 1 Comments:
Comments and Suggestions for Authors:
This a very interesting study and the findings add to the knowledge of heterologous vascular grafting. There is a very large amount of data and some of the table are difficult and hard to understand, specially when they showed no difference between both groups. it would be very helpful for the reader to show it in a more friendly manner.
Authors response:
Many thanks for these very supportive remarks. Regarding the generic point on the format of the tables, we agree that they are busy given the large amount of data included. On reflection, we feel they are ok, as additional changes might make them worse while removing data to simplify them would not be appropriate.
Reviewer 2 Report
Comments and Suggestions for Authors
This manuscript presents a well-designed pilot study assessing the feasibility, safety, and regenerative capacity of decellularized human saphenous veins (D-hSVs), with or without pre-seeding with porcine endothelial-like cells (pELCs), in a porcine carotid artery model. The topic is timely and relevant, addressing a critical need for improved vascular graft options. The methodology is sound, the study adheres to ethical standards, and the findings offer meaningful insights into vascular tissue engineering.
Specific Suggestions and Comments:
- Please include the age of the animals used. A summary table in the appendix with individual animal data (age, weight, and available physiological parameters) would strengthen the methodological transparency.
- On page 5, line 133, the differences in inner wall diameters are not clearly highlighted. Consider annotating statistical differences directly in the graph.
- The mechanisms underlying the CD31-positive endothelial cells in Figure 2 could be better explained, specifically, whether these are host-derived or residual seeded cells.
- A suggestion for future work: consider applying this model in disease conditions such as hypertension to explore translational relevance further.
- Please double-check all figure legends to ensure all abbreviations are defined consistently.
- A composite panel of elastin staining showing all analyzed vessel regions would enhance clarity; consider including this in the appendix.
- For improved readability, label histological images directly with group and condition (e.g., "seeded – patent," "non-seeded – occluded") as seen in Figure A7. Adding directional arrows or markers to highlight key features—such as thrombus in Figure A2 or cell types in Figure 2d—would improve interpretability.
- Review all histological panels carefully to ensure that the visual content clearly supports the text and figure legends. Some histology images (e.g., Figures 2 and 4) would benefit from higher resolution and annotated labels to aid interpretation.
- There are minor inconsistencies in formatting across tables and figures; standardizing fonts, labels, and spacing will improve overall presentation quality.
- While the seeded group (SD-hSV) exhibited slightly higher patency rates, the conclusion that cell seeding confers no benefit may be premature. Even limited early endothelial coverage could impact local immune responses, thrombogenicity, or vascular remodeling. A more comprehensive and mechanistic discussion of these potential effects would enhance the interpretation of the findings.
- The exclusive use of female pigs is understandable as a strategy to reduce variability. However, sex-based differences in vascular biology, immune response, and remodeling are well-documented and may affect outcomes. This limitation should be explicitly acknowledged in the discussion. Additionally, incorporating relevant literature on xenogeneic implantation outcomes in both sexes would provide a broader context.
- The manuscript would benefit from a deeper and more integrated discussion of the histological and vascular remodeling parameters analyzed. In particular, the relevance of observed changes in elastin, collagen, endothelial markers, and smooth muscle cell phenotypes to graft integration and function should be elaborated. Similarly, the interpretation of ultrasound data, including pulsatility index and diameter matching, should be more clearly connected to graft performance and clinical relevance. Strengthening these sections would provide a more complete understanding of the vascular adaptation process following graft implantation.
Author Response
Reviewer 2 Comments:
Comments and Suggestions for Authors:
This manuscript presents a well-designed pilot study assessing the feasibility, safety, and regenerative capacity of decellularized human saphenous veins (D-hSVs), with or without pre-seeding with porcine endothelial-like cells (pELCs), in a porcine carotid artery model. The topic is timely and relevant, addressing a critical need for improved vascular graft options. The methodology is sound, the study adheres to ethical standards, and the findings offer meaningful insights into vascular tissue engineering.
Many thanks for these very supportive remarks.
Specific Suggestions and Comments:
Please see all tracked changes related to revisions outlined below in word manuscript document.
-
Please include the age of the animals used. A summary table in the appendix with individual animal data (age, weight, and available physiological parameters) would strengthen the methodological transparency.
Many thanks for valuable comment. In the study we used 4 month old pigs. Data for weights at implant and termination are already included in the manuscript, but now with the addition of mention of ages and “Table A1” (Page 2, Lines 66-67: “Weight at 4-weeks was 82.9±2.6kg and 80.0±1.3kg in D-hSV and SD-hSV groups respectively (p=0.35), representing weight gains of 22.5±1.7% and 26.8±2.0% in the SD-hSV and D-hSV groups respectively (p=0.13) (Table A1)” and page 15, Line 385-386: “….female landrace large white pigs (n=14; weight 61.2±1.0 kg; age 3.9±0.0 months – Table A1)”…. Table A1 has been modified, as requested, to include ages and weights, alongside vein and graft information.
-
On page 5, line 133, the differences in inner wall diameters are not clearly highlighted. Consider annotating statistical differences directly in the graph.
Many thanks. We have now addressed this point by adding significance symbols to Figure 1 as shown in revised manuscript, and updated the figure legend accordingly (Page 6, Line 145).
In addition, we have updated the Statistical Analysis section to refer to posthoc tests being carried out (Page 15, Line 441).
-
The mechanisms underlying the CD31-positive endothelial cells in Figure 2 could be better explained, specifically, whether these are host-derived or residual seeded cells.
Many thanks. We have already covered these aspects in Discussion, page 11, lines 245 onwards. As we did not undertake GFP labelling on seeded cells we cannot be sure whether the CD31-positive endothelial cells observed in the seeded group were host-derived or residual cells. However, as clearly stated in the manuscript, non-seeded grafts had their lumen fully covered by CD31-positive endothelial cells, which can only be host-derived as either endothelial pre-cursors or migrated across the anastomosis. We have now revised the manuscript (page 16, lines 249-252) including a section stating that future work could use either GFP-labelling of seeded cells, or implanting ELC from male pigs into female pigs followed by fluorescent in situ hybridization might help expanding on this point.
-
A suggestion for future work: consider applying this model in disease conditions such as hypertension to explore translational relevance further.
Thank you for this remark. We are planning several future projects based on the findings of the present study. We have now clarified this in the discussion (page 13, lines 323-325)
-
Please double-check all figure legends to ensure all abbreviations are defined consistently.
Many thanks. We have checked for consistency all the figure legends and all abbreviations with relative spelling and are now happy with their content.
-
A composite panel of elastin staining showing all analyzed vessel regions would enhance clarity; consider including this in the appendix.
Many thanks. We have reported in full the results of elastin staining in Figure 5 and Table S4. We appreciate that creating yet another a composite panel with all elastin staining might help the reader to visualize the results. However, we feel that this additional panel to be included in the supplemental file will not add to the manuscript as the elastin results are already reported, while making the adding of a further figure to the already cumbersome submission. Hence, on reflection, we have elected to not include this additional composite panel.
-
For improved readability, label histological images directly with group and condition (e.g., "seeded – patent," "non-seeded – occluded") as seen in Figure A7. Adding directional arrows or markers to highlight key features—such as thrombus in Figure A2 or cell types in Figure 2d—would improve interpretability.
Many thanks for this valuable comments. We have now revised Figure A2 and A7, updated the figures legends accordingly.
-
Review all histological panels carefully to ensure that the visual content clearly supports the text and figure legends. Some histology images (e.g., Figures 2 and 4) would benefit from higher resolution and annotated labels to aid interpretation.
Many thanks. We have checked for consistency across all the histological panels and related legends, and in particular Figs 2&4, and all abbreviations with relative spelling, and are now happy with their content. We have added labels to highlight lumen, intimal thickening, media and acellular regions, where possible in figures 2 & 3, and updated figure legends accordingly.
-
There are minor inconsistencies in formatting across tables and figures; standardizing fonts, labels, and spacing will improve overall presentation quality.
Many thanks. We have now resolved the minor formatting inconsistencies in tables and figures.
We also picked up few typos in the text of the submitted manuscript, which have been resolved as shown with tracked changes of the revised submission.
-
While the seeded group (SD-hSV) exhibited slightly higher patency rates, the conclusion that cell seeding confers no benefit may be premature. Even limited early endothelial coverage could impact local immune responses, thrombogenicity, or vascular remodeling. A more comprehensive and mechanistic discussion of these potential effects would enhance the interpretation of the findings.
Thank you for this comment, however we respectfully disagree with this suggestion. Our data clearly confirm that 100% of the EC cells seen in the unseeded groups were from the host pigs. When comparing these unseeded grafts vs those seeded we see no evidence of more endothelial coverage in the seeded group vs the unseeded. Hence, our data strongly suggest that the pre-seeding had no beneficial effects. We have discussed this extensively in our Discussions, but have added to the text to make clearer (Page 12, Line 246-266: “….While we can be sure that 100% of ECs seen in the control D-hSV group were host cells, this cannot be confirmed for the SD-hSV group as those grafts had an average pre-seeding coverage of 48% before implant. Quint et al28 implanted in porcine carotid arteries either decellularised/nude tissue-engineered grafts or seeded with ECs from the intended porcine recipient with 65% lumen coverage. 3/3 of their nude grafts were occluded while 5/5 seeded grafts were patent at 30-day. In 4/5 of these patent grafts the endothelial coverage was made from 35% by seeded GFP labelled cells whereas the remaining 65% consisted of host cells. Our study differs markedly from the work by Quint and colleagues regarding graft preparation and this might explain the different outcome related to the nude graft, with our study showing >50% of nude grafts being patent at 30-day with extensive reendothelialization by host cells. Yet, both studies appear to indicate that pre-seeding with ECs may not be beneficial for vascular tissue engineering as most, if not all the cells present in the graft after in-vivo exposure appear to be host cells. This is in keeping with in relevant findings by others suggesting that monocytes seeded onto a mouse venous tissue engineered graft rapidly disappear within 2 hours after implant in-vivo29….”. Hence, on balance, we are happy with the way we have reported our results related to the effect of pre-seeding and how we have discussed them in relation to the current literature. Therefore, no changes are proposed.
-
The exclusive use of female pigs is understandable as a strategy to reduce variability. However, sex-based differences in vascular biology, immune response, and remodeling are well-documented and may affect outcomes. This limitation should be explicitly acknowledged in the discussion. Additionally, incorporating relevant literature on xenogeneic implantation outcomes in both sexes would provide a broader context.
Many thanks. We have now expanded that section (see page 13, lines 325-330) as follows.: “Finally, an additional limitation of our study is that we used only female pigs and therefore this study may not cover for sex-based differences in vascular biology, immune response and vascular remodelling typical of male pigs. However, we used female pigs only to minimise intra-experimental variability to increase the robustness of the findings while minimising animal husbandry issues associated with using a mixed gender cohort of animals”.
Regarding the xenogeneic implantation aspect of our study, we did not use immunosuppression as we implanted decellularised/acellular human vein grafts in pig. This approach is in keeping with the established clinical practice of implanting millions of commercial patches or artificial biological heart valves made from decellularised/acellular bovine pericardium in humans every year (Vahanian et al, (2022), 2021 ESC/EACTS Guidelines for the management of valvular heart disease, Eur Heart J). We have now included this text in our revised discussion (see page 11, lines 234-236, for the benefit of the readers.
-
The manuscript would benefit from a deeper and more integrated discussion of the histological and vascular remodeling parameters analyzed. In particular, the relevance of observed changes in elastin, collagen, endothelial markers, and smooth muscle cell phenotypes to graft integration and function should be elaborated. Similarly, the interpretation of ultrasound data, including pulsatility index and diameter matching, should be more clearly connected to graft performance and clinical relevance. Strengthening these sections would provide a more complete understanding of the vascular adaptation process following graft implantation.
Many thanks. We have checked our Discussion again after this generic comment. We had included >1.5 page of clear discussion in which our histological and USD findings are clearly discussed alone and within the context of the available literature. We have also interpreted or speculated on some of our data (e.g. USD data) when no relevant studies had been identified for comparison. On reflection, we feel that our discussion is balance enough and that expanding it in relation to the histological findings may make that part excessive while expanding it in relation to our USD will be too speculative due to lack of other similar studies in a similar animal model. Hence, while we have made quite few changes in our Discussion to address other points made by this reviewer, we have not taken on this comment for reasons explained above.